# Study on the Effects and Mechanism of Corilagin on A2780 Cell Apoptosis

**DOI:** 10.3390/cimb47020105

**Published:** 2025-02-07

**Authors:** Ziyang Xu, Yuhan Jiang, Tiantian Shan, Lei Hu, Minrui Wu, Hanxu Ji, Longjie Li, Yang Yi, Hongxun Wang, Limei Wang

**Affiliations:** 1College of Life Science and Technology, Wuhan Polytechnic University, Wuhan 430023, China; xuziyang20200@163.com (Z.X.); sweet@bio-sun.com.cn (T.S.); hu_lei@outlook.com (L.H.); wuminrui150@163.com (M.W.); j1752566276@163.com (H.J.); lilongjie@whpu.edu.cn (L.L.); wanghongxun7736@163.com (H.W.); 2College of Life and Science, Huazhong University of Science and Technology, Wuhan 430023, China; jyh1149310136@163.com; 3College of Food Science and Engineering, Wuhan Polytechnic University, Wuhan 430023, China; yiy86@whpu.edu.cn

**Keywords:** corilagin, ovarian cancer, apoptosis, mechanism, p53 signaling pathway

## Abstract

Previous studies have demonstrated corilagin’s inhibitory effects on the growth of various cancer cells. Given the limited research on corilagin’s impact on ovarian cancer, a particularly deadly gynecological malignancy, this study aimed to investigate corilagin’s influence on A2780 ovarian cancer cell apoptosis and its underlying mechanisms. The goal was to evaluate corilagin’s potential as a therapeutic agent for ovarian cancer. The results of the CCK-8 assay showed that corilagin inhibited the proliferation of A2780 ovarian cancer cells while exhibiting lower toxicity to normal ovarian surface epithelial cells (IOSE-80). We found that corilagin significantly altered the A2780 cell cycle, decreasing the proportion of cells in the G_0_/G_1_ and G_2_/M phases and inducing cell cycle arrest in the S phase. At low concentrations, corilagin induced apoptosis in A2780 cells, accompanied by a decline in mitochondrial membrane potential and calcium influx. Transcriptome sequencing analysis identified differentially expressed apoptosis-related genes in corilagin-treated A2780 cells, primarily within the PI3K-AKT pathway. Furthermore, qPCR and Western blot results confirmed the upregulation of *p53* and *Bax* genes and the downregulation of BCL-2. Corilagin also increased the expression of apoptotic factors caspase-9, caspase-3, PUMA, and cytochrome C, indicating its ability to induce apoptosis. Overall, corilagin effectively inhibited A2780 cell proliferation, induced cell cycle arrest, and triggered apoptosis. Its anti-tumor effect in vitro suggests its potential as a therapeutic agent for ovarian cancer A2780, especially through the PI3K/p53 pathway.

## 1. Introduction

Ovarian cancer, a malignant tumor of the ovary, is the most prevalent gynecological cancer among women and the fifth leading cause of death worldwide [1,2,3]. Over the years, numerous natural products have demonstrated therapeutic potential in patients with ovarian cancer. Notably, these natural components often exhibit minimal toxicity to healthy cells and tissues, making them a promising alternative to conventional medications for treating ovarian cancer. Corilagin(β-1-O-galloyl-3,6-(*R*)-hexahydroxydiphenoyl-D-glucose) is a natural plant polyphenol tannin compound found in various plants (Figure 1), including phyllanthus urinaria, geranium, orange grass, and white clover [4]. Additionally, it is present in plants known for their antioxidant and antiaging properties, such as emblica, longan, and olive [5,6]. Research has revealed corilagin’s diverse biological activities, including anti-tumor properties [7], cardiovascular disease treatment [8], antioxidant activity [9], and anti-inflammatory effects [10]. Due to its promising potential, corilagin has garnered significant attention from researchers, warranting further investigation and development.

Jia et al. [11] demonstrated that corilagin could inhibit the growth of ovarian cancer cells by blocking the transforming growth factor (TGF)-β signaling pathway. Their subsequent research revealed corilagin’s influence on the apoptosis pathway and its ability to sensitize epithelial ovarian cancer cells to chemotherapy by inhibiting the Snail-glycolysis pathway [12]. Rukset Attar et al. [13] discovered that corilagin could stimulate the MAPK and phosphatidylinositol signaling systems, suggesting its anti-tumor effects on downstream pathways. These findings collectively indicate corilagin’s potential as a novel therapeutic approach to epithelial ovarian cancer.

Interestingly, Josianne Rocha Barboza et al. [14] reported that Difeng Gum extract, containing corilagin components, regulates the cell cycle by activating caspase-3 and PARP protein cleavage. This activation ultimately inducing apoptosis in A2780 ovarian cancer cells. While these findings suggest corilagin’s apoptotic effects on A2780 cells, the precise mechanisms underlying its action remain unclear. Our study aimed to detect changes in the inhibitory rate of A2780 human ovarian cancer cells under different drug concentrations by the CCK8 method. We also investigated how corilagin inhibits A2780 cell activity by employing flow cytometry to analyze cell cycle, apoptosis, mitochondrial membrane potential, and intracellular calcium concentration. Furthermore, we explored the molecular mechanisms involved in corilagin-induced A2780 cell apoptosis through transcriptome gene sequencing and real-time fluorescence qPCR, which was further verified by Western blotting.

## 2. Materials and Methods

### 2.1. Materials and Reagents

Corilagin (purity ≥ 98%) was purchased from Tongtian Biology, Shanghai, China. Cell culture materials, including T25 flasks, 96- and 6-well plates, cell cryopreservation tubes, and 24-well 8 μm transwell chambers, were purchased from Corning, CA, USA. Matrigel was acquired from Becton, Dickinson Company, Franklin Lakes, NJ, USA. Additionally, Cell Counting Kit-8 (CCK-8) was provided by Tongren, China. 5-fluorouracil (5-FU) was acquired from Sigma, St. Louis, MO, USA. The T25 flasks, 96-well plates, and 6-well plates were purchased from Corning, Glendale, AZ, USA. Trypsin was obtained from Gino Bio-medical Technology Company, Hangzhou, China, and anhydrous ethanol, n-butanol, chloroform, and methanol from Sinopharm Chemical Reagent, Shanghai, China. The 40–60 μm silica gel was obtained from Qingdao Ocean Chemical, Qingdao, China. The real-time PCR kit, Sephadex LH-20, was obtained from BioRad, Hercules, CA, USA. The total RNA extraction kit from Omega, GA, USA, and the reverse transcription kit from Takara, Shiga, Japan. The cell cycle detection kit, the mitochondrial membrane potential detection kit, and the intracellular calcium ion concentration detection kit were obtained from Beyotime Biotechnology, Shanghai, China. The apoptosis assay kit was purchased from BestBio, Shanghai, China.

### 2.2. Cell Culture

The human ovarian cancer cell line A2780 was obtained from the Shanghai Institute of Cell Research, Chinese Academy of Sciences, Shanghai, China. The normal ovarian epithelial cell line IOSE-80 was obtained from Shanghai SIG Biotechnology, Shanghai, China. Cells were cultured in DMEM medium supplemented with 5% heat-inactivated FBS, 100 U mL^−1^ of penicillin, and 0.1 mg mL^−1^ of streptomycin. Cells were cultured in an incubator (37 °C, 5% CO_2_). Cells were grown to about 80% confluence in a 1:3 ratio of passaged culture.

### 2.3. Cell Proliferation Assay

A Cell Counting Kit-8 was used to measure cell viability. Briefly, cancer cells (2 × 103 cells per well in 100 μL medium) and normal ovarian epithelial cells (4 × 103 cells per well in 100 μL medium) were seeded in 96-well plates. The next day, the medium was aspirated, and equal amounts of corilagin and 5-fluorouracil (20, 40, 60, 80, and 100 μmol/mL) were added, followed by incubation for 24 h or 48 h. The media was aspirated out and the wells were washed once with 1 × phosphate-buffered saline (PBS), followed by 100 µL of fresh media and 10 μL of CCK-8. The plate was then incubated for 4 h at 37 °C. The absorbance of the formazan products was measured at 450 nm using a microplate reader. The cell inhibition rate was calculated by dividing the number of viable cells in the compound-treated groups by that in the control group, using the following equation:Cell inhibition rate (%) = [1 − (OD_test group_ − OD_blank group_)/(OD_control group_ − OD_blank group_)] × 100 (1)

The half maximal inhibitory concentration (IC_50_) was then calculated and defined as the concentration accounting for 50% of viability.

### 2.4. Cell Cycle Assay

A commercial cell cycle detection kit was used to detect cell cycle distribution. Briefly, a cell suspension in the logarithmic growth phase, with a concentration of 1 × 10^6^ cells/mL, was inoculated into a 6-well plate. Each well received 1 mL of the suspension and was cultured for 24 h. The culture medium was then replaced with 2 mL of a solution containing varying concentrations of corilagin (20, 40, 60, 80, and 100 μmol/mL). Cells were further cultured for 48 h in a 37 °C incubator. Cells were trypsinized for 30 s, centrifuged at 1500 rpm for 5 min at 4 °C, and washed twice in cold PBS. Precooled 75% ethanol was added, and the mixture was stored overnight at 4 °C in the dark for fixation. After centrifugation, 200 μL of RNase was added and incubated for 30 min at 37 °C in the dark. Finally, 100 μL of propidium iodide (PI) staining solution (100 µg/mL) was added for 30 min at 4 °C in the dark, and cell analysis was performed on a flow cytometer (BD FACS Calibur) within 1 h. The recorded cell fluorescence was used to analyze cell cycle distribution.

### 2.5. Apoptosis Detection

The apoptosis of A2780 cells was detected using the Annexin V-FITC/PI Apoptosis Kit. Briefly, a cell suspension in the logarithmic growth phase was inoculated into a 6-well plate at a concentration of 1 × 10^5^ cells/mL. Each well received 1 mL of the suspension and was cultured for 24 h. Subsequently, 1 mL of a solution containing varying concentrations of corilagin and 5-FU (20, 40, 60, 80, and 100 μmol/mL) was added to each well, and the cultures were incubated at 37 °C for an additional 48 h. Following a 30-s trypsin digestion, the suspended cells were harvested by centrifugation after being dispersed. The cells were washed twice with ice-cold PBS, and the PBS was discarded. The Annexin V binding solution was adjusted to contain 5 × 10^4^ cells/mL. Five microliters of Annexin-V-FITC staining solution were added and incubated for 15 min at 4 °C in the dark. Finally, 10 μL of PI staining solution was added, followed by a 5-min incubation at 4 °C in the dark. Finally, cells were observed via flow cytometry (BD FACS Calibur) and the apoptosis rate analyzed Flowjo V10 software.

### 2.6. Assessment of the Mitochondrial Membrane Potential

The lipophilic fluorochrome JC-1 (5,5′,6,6′-tetrachloro-1,1′,3,3′-tetraethylbenzimidazolcarbocyanine iodide) was used to assess changes in mitochondrial membrane potential. Briefly, a 6-well plate was inoculated with a cell suspension in the logarithmic growth phase at 1 × 10^5^ cells/mL. Each well received 1 mL of the suspension and was cultured for 24 h. The culture medium was then replaced with fresh medium containing varying concentrations of corilagin (20, 40, 60, 80, and 100 μmol/mL). After adding 2 mL of solution, the cultures were incubated at 37 °C for 48 h. Cells were trypsinized for 30 s, centrifuged at 1500 rpm for 5 min at 4 °C, and collected. Following two washes with pre-cooled PBS, the cells were resuspended in 0.5 mL of JC-1 staining working solution and incubated at 37 °C for 20 min. After two additional washes with JC-1 Staining Buffer, the cell suspension was adjusted to a concentration of 5 × 10^4^ cells/mL and analyzed by flow cytometry (BD FACS Calibur) within 1 h.

### 2.7. Intracellular Calcium Ion Concentration Assay

The calcium ion concentration in A2780 cells was determined using a Fluo-3 AM calcium concentration detection kit. Briefly, a logarithmic growth phase cell suspension (1 × 10^5^ cells/mL) was inoculated into a 6-well plate at a density of 1 mL/well. After a 24-h incubation period, the culture medium was replaced with 2 mL of medium containing various concentrations of corilagin (20, 40, 60, 80, and 100 μmol/mL). Subsequently, the cultures were maintained in a constant-temperature incubator at 37 °C for 48 h. Following trypsin digestion and centrifugation, the cells were collected and washed twice with Hanks’ Balanced Salt Solution (HBSS). The cell concentration was then adjusted to 5 × 10^4^ cells/mL. BBcellProbeTM F3 staining solution was added and incubated at 37 °C for 20 min. After two to three washes with HBSS, the cells were resuspended, incubated at 37 °C for 10 min, and analyzed using a flow cytometer (BD FACS Calibur) within 1 h.

### 2.8. Transcriptomic Data Analysis

A 1 × 10^5^/mL cell suspension was inoculated in the logarithmic growth phase into a 6-well plate, 1 mL per well, and 60 μmol/mL corilagin was added. The culture was continued in a 37 °C constant-temperature incubator for 24 h. Then, the cells were trypsinized for 30 s and centrifuged at 1500 rpm for 5 min at 4 °C to collect the cells. The cells were washed twice with precooled PBS, and the PBS was discarded. Subsequently, the cells were collected and stored in liquid nitrogen. Six A2780 cell samples were included, with three in the blank group and three in the 60 μmol/mL corilagin-treated group.

The TRIzol method was used to extract total RNA from samples of the treatment group and blank group. Raw image files were obtained through the experimental platform’s high-throughput sequencing. The preliminary experimental data underwent base recognition and error filtering to produce raw sequencing fragments (Raw Reads). The differential gene expression between the treatment group and the control group was analyzed using Cuffdiff-2.2.1 software. The differential expression fold change was calculated using FPKM values (FPKM represents fragments per kilobase and per million). The gene background of the treatment group was compared to the blank group to screen for differentially expressed genes (DEGs). The functional enrichment analysis of DEGs in gene ontology (GO) and Kyoto Encyclopedia of Genes and Genomes (KEGG) pathway database was applied using phyper in the R package.

### 2.9. Real-Time Fluorescent Quantitative PCR

This study employed real-time fluorescence quantitative PCR to analyze gene expression levels within predicted pathways identified through transcriptome sequencing. Specifically, A2780 cells were treated with 60 µM corilagin for 24 h. Total RNA was extracted with a TRIzol reagent according to the manufacturer’s instructions. The RNA was reverse-transcribed using a reverse transcription kit to obtain cDNA. The synthesized cDNA was used as a template for subsequent PCR reactions. In simple terms, the 20 μL reaction mixture consists of 2 μL cDNA, 10 μL SYBRs Premix Ex TagTM, and 8 μL specific target primers (10 μM). All reactions were performed on a CFX96 real-time PCR System (BioRad, Hercules, CA, USA). The thermal cycling settings for qPCR were 95 °C for 30 s, followed by 40 cycles at 95 °C for 5 s and 60 °C for 30 s. β-Actin was used as the internal control. Relative gene expression levels were calculated by the 2^−ΔΔCt^ method. The primers used for qPCR are presented in Table 1.

### 2.10. Western Blot Analysis

A 1 × 10^5^/mL cell suspension was inoculated in the logarithmic growth phase into a 6-well plate, 1 mL per well, and 60 μmol/mL corilagin was added. The culture was continued in a 37 °C constant-temperature incubator for 24 h. A2780 cells were treated with 60 µM corilagin for 24 h. The cells were digested with trypsin, washed twice with 1 mL of PBS, and transferred to a 1.5 mL centrifuge tube. Subsequently, 100 mL of cell lysate (10 mm of Tris-HCl, pH 8.0, 1 mm of EDTA, 20% SDS, 5 mm of DTT, 10 mm of PMSF) was obtained by lysing the cells on ice for 30 min and sonicating 10 times for 2 s each time. The cell suspension was centrifuged at 12,000 rpm and 4 °C for 5 min. The protein concentration was determined using the BCA kit, and samples were resolved using electrophoresis. In addition, a quarter of the sample volume of the protein-loaded buffer was added to the protein sample. The protein was denatured at 95 °C for 5 min and centrifuged at 12,000 rpm and 4 °C for 5 min after cooling at room temperature. The protein samples were transferred to a 10% SDS-polyacrylamide gel for electrophoretic separation. The samples were then incubated in boiling water for 5 min. The samples were electrophoretically separated in 12.5% SDS-PAGE gel at constant pressure (120 V) and then transferred to a polyvinylidene fluoride membrane (Millipore, Burlington, MA, USA). The following primary antibodies were used to block the membranes with 5% milk and incubate them overnight at 4 °C: anti-pIRS1(Ser307) (1:1000; AI623-1, Beyotime, China) and anti-IRS1 (1:1000; 2382, CST, Danvers, MA, USA), while beta-actin was chosen as an internal control. Quantity One software V4.6.6 (Bio-Rad, Hercules, CA, USA) was used to quantitatively analyze the optical densities of the bands of interest.

### 2.11. Statistical Analysis

All statistical analyses were performed using GraphPad Prism 8.0.2 software and all data are expressed as the mean ± SD of three independent experiments. For all experiments, a *p*-value < 0.05 was considered statistically significant.

## 3. Results

### 3.1. Effect of Corilagin on the Viability of Ovarian Cancer A2780 Cells

Figure 2 shows that corilagin’s proliferation inhibition rate on A2780 cells was concentration-dependent, and the inhibition rate was positively correlated with drug concentration. The results showed that the treatment effect of 5-FU was better than that of corilagin under experimental conditions of 20–60 μmol/mL for 24 h and 48 h. However, under experimental conditions of 80–100 μmol/mL for 24 h and 48 h, the effects of corilagin and 5-FU gradually converged. The semi-inhibitory concentration (IC_50_ value) was calculated based on the strength of the inhibition rate (Table 2). Results showed that the IC_50_ values of corilagin-treated A2780 cells for 24 h and 48 h were 61.32 and 47.81, respectively, whereas the IC_50_ values of 5-FU-treated A2780 cells for 24 h and 48 h were 36.46 and 22.81, respectively. In addition, corilagin was much less cytotoxic in normal ovarian epithelial cells, with IC50 values of 263.72 and 174.52 at 24 h and 48 h, respectively. Based on the IC_50_ results, we found that the longer the treatment time, the stronger the killing effect on A2780. Collectively, these results suggest that corilagin inhibited the proliferation of ovarian cancer A2780 cells and showed a good dose-effect relationship.

### 3.2. Effect of Corilagin on the A2780 Cell Cycle in Ovarian Cancer

The cell cycle is the fundamental process governing cellular life. In tumor cells, aberrant regulation of the cell cycle results in uncontrolled proliferation. Interfering with the cell cycle of tumor cells at specific stages can impede tumor growth. Previous research has shown that corilagin can influence the cell cycle of tumor cells [15,16]. As shown in Figure 3, corilagin administration at a concentration of 20 μmol/mL caused a significant alteration in cell cycle distribution compared to the control group. Notably, there was a marked decrease (*p* < 0.05) in the proportion of cells in the G_2_/M and G_0_/G_1_ phases, with a concomitant increase in the S phase. As the concentration of corilagin was further elevated (20, 40, 60, 80, and 100 μmol/mL), the trend persisted, with a continued reduction in the proportion of cells in the G_0_/G_1_ and G_2_/M phases and a corresponding rise in the S phase. These findings collectively suggest that low-dose corilagin can effectively reduce the proportion of cells in the G_2_/M and G_0_/G_1_ phases in A2780 cells, leading to cell cycle arrest in the S phase.

### 3.3. Effect of Corilagin on Apoptosis of Ovarian Cancer A2780 Cells

Previous research established a strong correlation between the inhibition of tumor cell cycle progression and the induction of apoptosis [17]. The current investigation sought to explore this connection further. A2780 cells were exposed to varying corilagin concentrations for 48 h. As illustrated in Figure 4, a gradual increase in the percentage of apoptotic cells was observed following treatment. These results suggest a concentration-dependent effect of corilagin on A2780 cell apoptosis. A significant increase in apoptosis (*p* < 0.05) was observed at a concentration of 20 μmol/mL, with an associated apoptosis rate of 13.03%. Furthermore, a progressive rise in both the total apoptosis rate and the rates of early and late cell apoptosis was evident with increasing corilagin concentrations (20, 40, 60, 80, and 100 μmol/mL). Notably, the highest concentration tested (100 μmol/mL) yielded a substantial cell apoptosis rate of 53.37%. When the corilagin concentration was lower than 60 μmol/mL, its apoptosis-inducing effect on A2780 cells was slightly weaker than that of the positive drug 5-FU. When the corilagin concentration reached 80–100 μmol/mL, the apoptosis-inducing effect of A2780 cells was not significantly different from that of the positive drug 5-FU. In conclusion, the present study provides compelling evidence that corilagin significant influences apoptosis induction in A2780 cells.

### 3.4. Effect of Corilagin on the Membrane Potential of Mitochondria in Ovarian Cancer A2780 Cells

Mitochondria play a crucial role in cell apoptosis, acting as the “switch” that initiates apoptosis [18]. Current evidence suggests that during apoptosis, the permeability of the mitochondrial membrane increases while the cell membrane potential decreases. In the present study, as the concentration of corilagin increased (from 20, 40, 60, 80, to 100 μmol/mL), the red/green fluorescence ratio gradually decreased, and the decrease in the ratio of mitochondrial transmembrane potential in cells correspondingly increased. When the concentration reached 100 μmol/mL, the mitochondrial transmembrane potential decreased by 23.26% ± 1.65%. This trend indicated that as the concentration of the compound increased, the degree of apoptosis in A2780 cells also increased. These findings demonstrate that corilagin significantly affects the mitochondrial membrane potential of A2780 cells in a concentration-dependent manner. At low doses, it could effectively reduce the mitochondrial membrane potential of A2780 cells and induce apoptosis (Figure 5).

### 3.5. Effect of Corilagin on Intracellular Calcium Ion Concentration in Ovarian Cancer A2780 Cells

Calcium ions (Ca²⁺), serving as a pivotal second messenger within cells, are predominantly sequestered within the endoplasmic reticulum. Upon the initiation of cell apoptosis, Ca²⁺ effluxes from the endoplasmic reticulum into the intracellular milieu, activating apoptotic factors such as caspases, which ultimately culminate in programmed cell death [19]. Figure 6 depicts the progressive augmentation of intracellular calcium ion levels as drug concentrations rise after 48 h of corilagin treatment. When the corilagin concentration attained 100 μmol/mL, a notable 32.94% ± 1.22% increase in intracellular calcium ion concentration was observed, signifying a concomitant elevation in the apoptotic rate of A2780 cells. These findings collectively demonstrate that corilagin exerts a substantial influence on the intracellular calcium ion concentration within A2780 cells; however, this effect is contingent upon the drug concentration. Even at low dosages, corilagin can significantly elevate intracellular calcium ion levels, which aligns with the observed alterations in mitochondrial membrane potential, as illustrated in Figure 6. 

### 3.6. Effect of Corilagin on A2780 Cell Transcriptome Gene Sequencing

The experiment employed high-throughput sequencing to investigate the molecular mechanism of drug action by examining alterations in cellular genes following drug administration. Transcriptome gene sequencing revealed 792 DEGs in A2780 cells exposed to corilagin, comprising 467 upregulated genes and 325 downregulated genes. Gene ontology (GO) analysis indicated that these DEGs were primarily enriched in biological processes such as cellular processes and biological regulation, cellular components such as organelles and membrane components, and molecular functions such as catalytic activity and binding (Figure 7). Subsequent GO-C enrichment analysis and KEGG pathway enrichment analysis of differentially expressed genes identified a significant convergence of apoptosis-related DEGs within the PI3K-AKT signaling pathway (Figure 8). Consequently, it is highly conceivable that the alterations in specific genes within the PI3K-AKT signaling pathway of A2780 cells induced by corilagin are responsible for triggering apoptosis and modifying the cell cycle.

### 3.7. Effect of Corilagin on Gene Expression in Ovarian Cancer A2780 Cells

Figure 9 illustrates the expression of PI3K-AKT signaling pathway-related genes in A2780 cells following treatment with 60 μmol/mL corilagin for 24 h. Compared to the control group, a notable increase in *p53* gene expression was observed, accompanied by a significant upregulation of downstream apoptotic genes *PUMA, PERP, CASPASE9, CASPASE3, CYTOCHROME C*, and *BAX* (*p* < 0.05). Conversely, the expression of the *BCL-2* gene was significantly reduced, indicating the activation of apoptosis-related genes and the subsequent induction of apoptosis. Additionally, treatment with corilagin resulted in a significant upregulation of the *p21* gene and a significant downregulation of the *CYCLIN* gene in A2780 cells compared to the control group, suggesting corilagin’s impact on the cell cycle (*p* < 0.05).

### 3.8. Effect of Corilagin on Protein Expression in Ovarian Cancer A2780 Cells

Western blot analysis (Figure 10A) showed that p21 protein content involved in cell cycle arrest was increased after treatment of A2780 cells with corilagin compared to the control group (*p* < 0.05). After treating A2780 cells with corilagin for 24 h, the caspase-3 protein was cleaved and its corresponding splicing content increased. The caspase-9 content also increased significantly (*p* < 0.01), indicating that the caspase protein contributes to the corilagin-induced apoptosis of A2780 cells. Western blot was used to detect changes in the expression levels of Bcl-2 family-related proteins. The results showed that the expression of pro-apoptotic protein BAX was upregulated, whereas anti-apoptotic protein BCL-2 was downregulated. In addition, p53 protein up-regulation during corilagin-induced apoptosis can increase downstream PUMA protein expression, alter MMP, activate pro-apoptotic BCL-2 family proteins, and release Cytochrome C to induce apoptosis (Figure 10B).

## 4. Discussion

Research investigating the anti-cancer potential of corilagin was initiated in 1985. Within RNA tumor viruses, it was observed to inhibit reverse transcriptase activity [20]. Tong et al. [21] reported that corilagin stimulated autophagy and reactive oxygen species-mediated apoptosis in breast cancer cell lines (MDA-MB-231 and MCF-7). Deng et al. [22] observed a concentration-dependent downregulation of p-AKT expression and upregulation of p53 protein expression in the SMMC-7721 cell line following treatment with corilagin. Additionally, the breakdown of caspase-9, caspase-3, and PARP was detected, supporting the stimulation of the intrinsic apoptotic pathway. While the above studies overlap in their assertion of corilagin’s efficacy in inhibiting tumor activity, its specific effects on ovarian cancer and the underlying mechanisms have yet to be thoroughly evaluated [23,24]. This study sought to elucidate the effects of corilagin on the proliferation and apoptosis of ovarian cancer cells. Through transcriptome sequencing analysis and qPCR analysis, the relationship between its mechanism of action and the PI3K Akt signaling pathway was explored. The above results were verified by Western blot.

The cell cycle serves as the foundation for cell proliferation, primarily emphasizing two key events: the duplication of genomic DNA during the S phase and its subsequent distribution to two daughter cells. In the event of a disruption to the process of DNA replication, a series of DNA repair signaling events will be initiated, resulting in the extension of the S phase [25]. Cells are arrested in the S phase of the cell cycle for DNA damage. Cells arrested in the S phase are more likely to undergo apoptotic cell death through p53-dependent and p53-independent pathways. Blocking tumor cells at a certain phase of the cycle is one of the mechanisms to inhibit further tumor development. We established an in vitro ovarian cancer model using A2780 cells. Flow cytometry analysis revealed that corilagin significantly influenced the cell cycle of A2780 cells. Low doses of corilagin effectively reduced the abundance of A2780 cells in the G_2_/M and G_0_/G_1_ phases, resulting in their accumulation in the S phase. Increasing consensus suggests that the *p21* gene, a downstream target of *p53*, plays a crucial role in the cell cycle process [26,27,28]. The expression of *p21* regulates the activity of cyclin-dependent protein kinase (CDK), thereby influencing the cell cycle and controlling cell proliferation [29]. Western blot analysis showed that corilagin can lead to the accumulation and activation of p21. In summary, corilagin can upregulate the expression of the *p21* gene, inhibit A2780 cells in the S phase, and lead to cell cycle arrest and inhibition of cell proliferation.

Apoptosis is an autonomous and orderly cell death process regulated by genes under specific physiological or pathological conditions. This process is essential for maintaining the stability of the in vivo organic environment [30,31,32]. In this study, flow cytometry results demonstrated that corilagin induces cell apoptosis. As the concentration of corilagin increased, the overall apoptosis rate, as well as the percentage of early and late apoptotic cells, also increased. These findings align with changes in mitochondrial membrane potential (MMP) and Ca^2+^ concentrations. Transcriptome gene analysis revealed a significant concentration of apoptosis-related genes within the PI3K-AKT pathway following corilagin treatment of A2780 cells, suggesting corilagin’s ability to induce cell apoptosis through this pathway. To further explore the relationship between corilagin’s inhibition of cell apoptosis and the p53 signaling pathway, we initially analyzed the alterations in related genes and proteins within this pathway using qRT-PCR and Western blot. The results indicated an upregulation of the *p53* gene in cells exposed to corilagin, leading to a stimulated increase in the expression of the downstream gene *Bax*. This increase in *Bax* expression, coupled with a reduction in *BCL-2*, facilitated the onset of cell apoptosis. Additionally, the upregulated expression of *CASPASE9, CASPASE3, PUMA*, and *CYTOCHROME C* apoptosis factors suggests that corilagin can induce cell apoptosis through the PI3K/p53 pathway.

In summary, this study demonstrates that corilagin’s in vitro anti-tumor effects are primarily mediated through two distinct mechanisms. On the one hand, corilagin regulates the expression of p21-related genes in A2780 cells, leading to alterations in the cell cycle. On the other hand, it upregulates the expression of *p53, PUMA, BAX, CASPASE9, CASPASE3*, and *CYTOCHROME C* genes, triggering apoptosis in A2780 cells (Figure 11). It is worth noting that corilagin’s anti-proliferative activity against ovarian cancer A2780 is not very strong. Studies have also shown that when the two drugs affect cancer cells through the same pathway or act on the same target, the effect of the combination is often greater than or equal to the effect of the drug alone [33]. Therefore, the combination of corilagin with other clinical agents targeting the PI3K/p53 pathway is an effective therapeutic strategy [7]. Moreover, the low bioavailability of corilagin and its metabolites poses a significant challenge. Therefore, exploring existing and innovative drug delivery systems to ensure the delivery of effective drug concentrations to the body and achieving therapeutic objectives is crucial for addressing this issue [34,35,36]. Designing a more optimized drug delivery system to facilitate the internal delivery of corilagin is a promising future research direction. In conclusion, this study offers valuable insights into the treatment of ovarian cancer and provides references for the development of related health foods.

## 5. Conclusions

(1) Corilagin has good inhibitory effects on ovarian cancer A2780 cells while being less toxic to normal ovarian epithelial cells IOSE-80. It can affect the cell cycle of A2780, notably reducing the G_0_/G_1_ and G_2_/M phase cell ratios and blocking cells in the S phase. Exposure to low concentrations of corilagin effectively induces apoptosis, decreases mitochondrial transmembrane potential, and promotes calcium ion influx in A2780 cells;

(2) Transcriptome gene analysis revealed a significant concentration of apoptosis-related genes within the PI3K-AKT pathway following corilagin treatment of A2780 cells, suggesting corilagin’s potential for inducing cell apoptosis through this pathway. The qPCR and Western blot results confirmed that incubating A2780 cells with 60 μmol/mL of corilagin for 24 h resulted in an upregulation of the *p21* gene, a downregulation of *Cyclin D*, and alterations in the cell cycle. By contrast, the upregulation of *p53, PUMA, BAX, CASPASE9, CASPASE3*, and *CYTOCHROME C* gene expressions induced cell apoptosis.

## Figures and Tables

**Figure 1 cimb-47-00105-f001:**
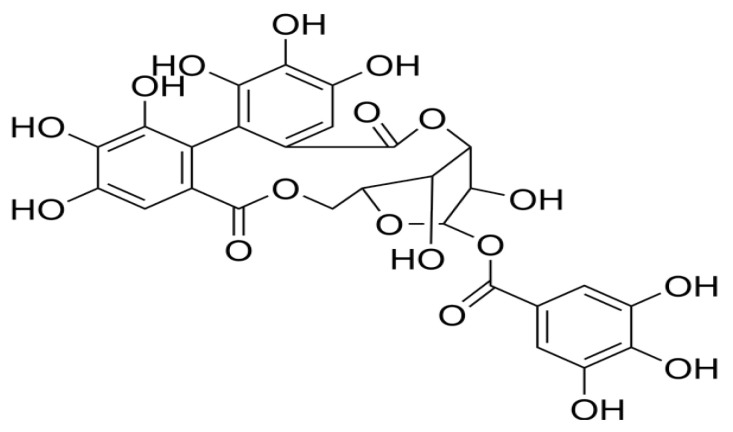
The chemical structure of corilagin.

**Figure 2 cimb-47-00105-f002:**
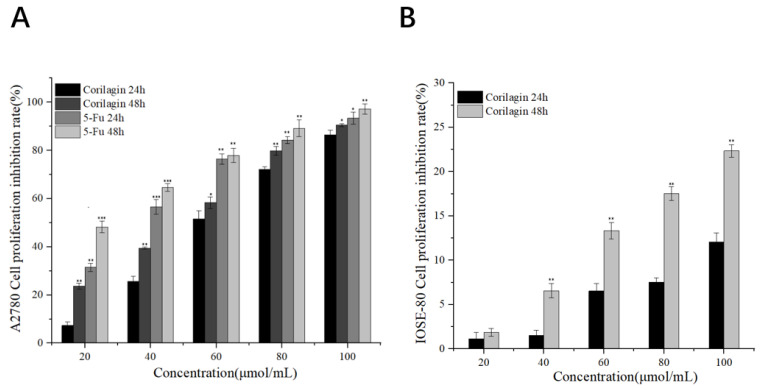
Effect of corilagin on the proliferation of A2780 and IOSE-80 cells. (**A**) A2780 cells were treated with different concentrations of corilagin and 5-FU (20, 40, 60, 80, and 100 μmol/mL) for 24 and 48 h, respectively, and detected by CCK-8 assay. (**B**) IOSE-80 cells were treated with different concentrations of corilagin (20, 40, 60, 80, and 100 μmol/mL) for 24 and 48 h and detected by CCK-8 assay. Data are means ± SDs (n = 3). * *p* < 0.05, ** *p* < 0.01, *** *p* < 0.001 vs. corilagin 24 h.

**Figure 3 cimb-47-00105-f003:**
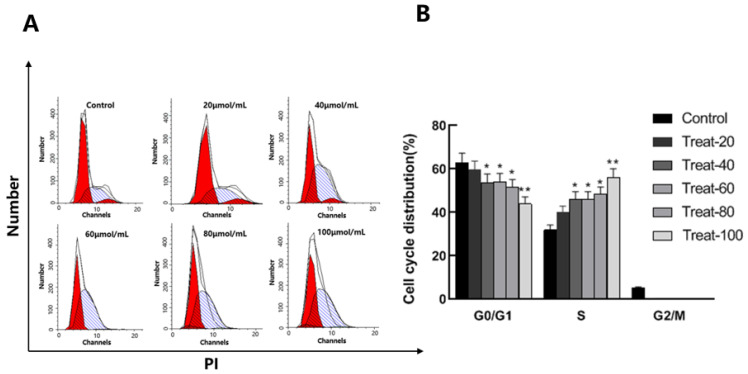
Effects of corilagin on the A2780 cell cycle. (**A**) Intracellular fluorescence intensity of A2780 cells cultured with corilagin (0, 20, 40, 60, 80, 100 μmol/mL) for 48 h. (**B**) Average fluorescence intensity of A2780 cells. Data are means ± SDs (n = 3). * *p* < 0.05, ** *p* < 0.01 vs. 0 μmol/mL.

**Figure 4 cimb-47-00105-f004:**
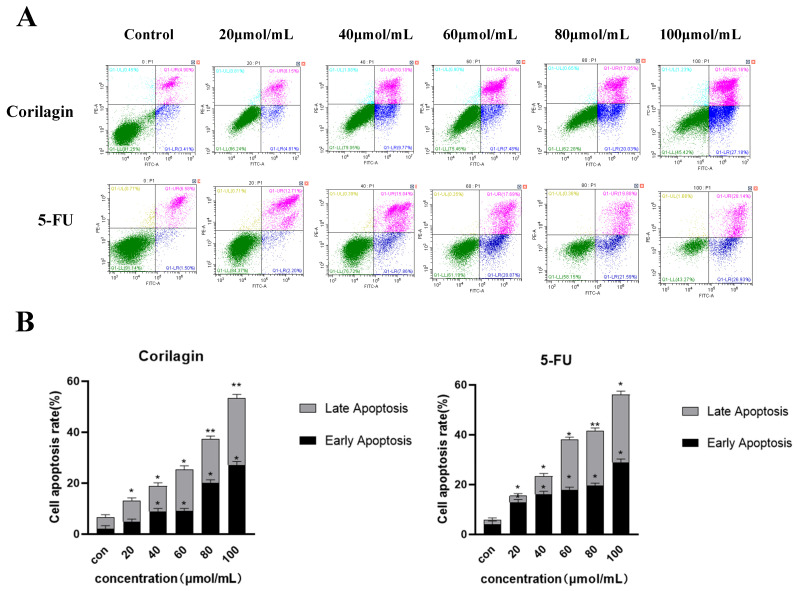
Effects of corilagin on the apoptosis of A2780 cells. (**A**) FITC-Annexin V/PI double-staining flow cytometry showing the apoptosis rate of A2780 cells after treatment with corilagin and 5-FU (0, 20, 40, 60, 80, 100 μmol/mL) for 48 h. (**B**) Average apoptosis rate of A2780 cells. PI, propidium iodide. Data are means ± SDs (n = 3). * *p* < 0.05, ** *p* < 0.01 vs. 0 μmol/mL.

**Figure 5 cimb-47-00105-f005:**
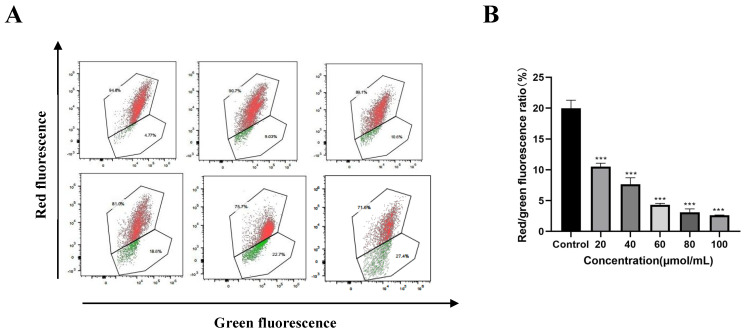
Effects of corilagin on mitochondrial membrane potential in A2780 cells. A2780 cells were incubated with different concentrations of corilagin for 48 h. ∆ψm was evaluated using JC-1 in treated cells. (**A**) ∆ψm after treatment of cells with 0, 20, 40, 60, 80, 100 μmol/mL corilagin. (**B**) JC−1 red/green fluorescence ratio. Data are means ± SDs (*n* = 3). *** *p* < 0.001 compared with control.

**Figure 6 cimb-47-00105-f006:**
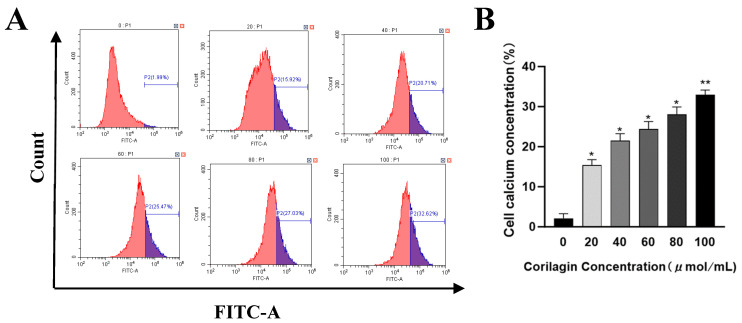
Effect of corilagin on intracellular Ca²⁺ concentrations in A2780 cells. A2780 cells were incubated with different concentrations of corilagin for 48 h. A Fluo 3-AM assay was performed to determine Ca²⁺ concentration. (**A**) Intracellular Ca²⁺ concentration after treatment of cells with 0, 20, 40, 60, 80, and 100 μmol/mL. (**B**) average cytoplasmic calcium rate of A2780 cells. Data are means ± SDs (*n* = 3). * *p* < 0.05, ** *p* < 0.01 vs. 0 μmol/mL.

**Figure 7 cimb-47-00105-f007:**
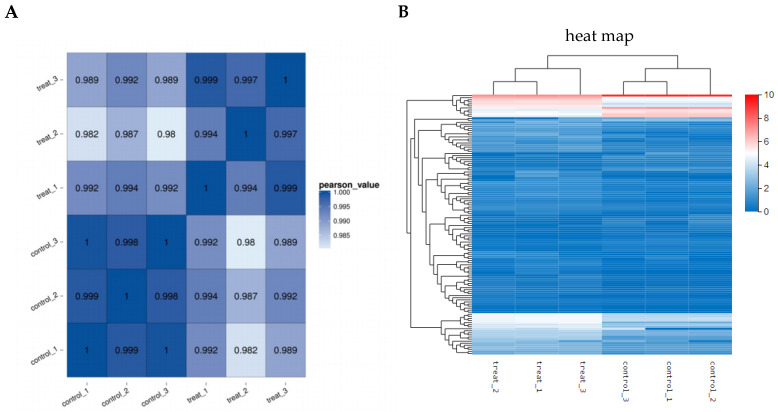
Distribution of differential gene expression. (**A**) Sample correlation analysis. (**B**) Analysis of gene differential expression among samples. (**C**) GO-cfp classification of differential gene function.

**Figure 8 cimb-47-00105-f008:**
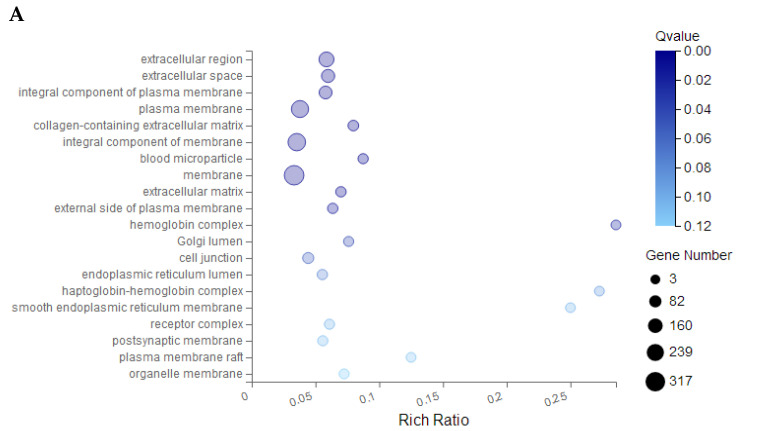
GO and KEGG pathway analysis of hub genes. (**A**) GO analysis. (**B**) KEGG pathway analysis. GO, gene ontology; KEGG, Kyoto encyclopedia of genes and genomes.

**Figure 9 cimb-47-00105-f009:**
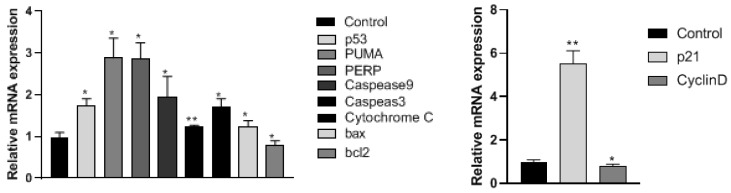
qPCR testing gene changes in A2780 cell, Data are means ± SDs (*n* = 3). * *p* < 0.05; ** *p* < 0.01.

**Figure 10 cimb-47-00105-f010:**
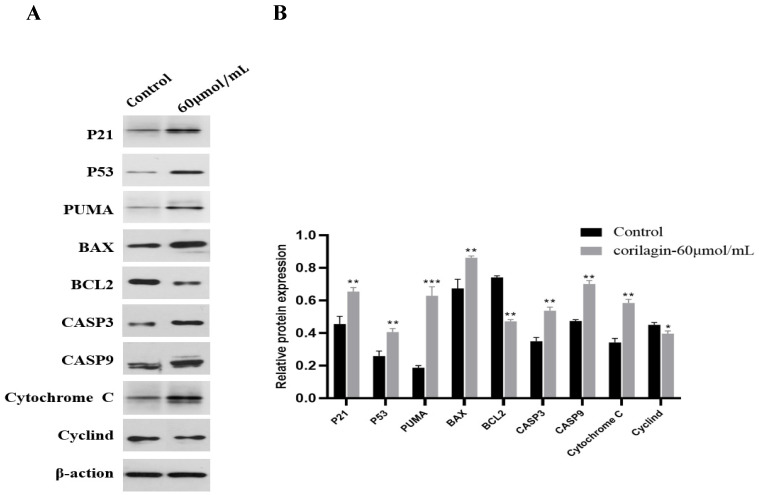
The expression levels of apoptosis-related proteins in A2780 cells. (**A**) Western blot analysis of cytochrome C, p53, Bax, Bcl-2, p21, PUMA, caspase-9, Cyclind, and caspase-3 expression in A2780 cells. Cells were treated with corilagin at 60 μmol/mL for 24 h. (**B**) The average protein band is in grayscale. Data are means ± SDs (*n* = 3). * *p* < 0.05, ** *p* < 0.01, *** *p* < 0.001 vs. control.

**Figure 11 cimb-47-00105-f011:**
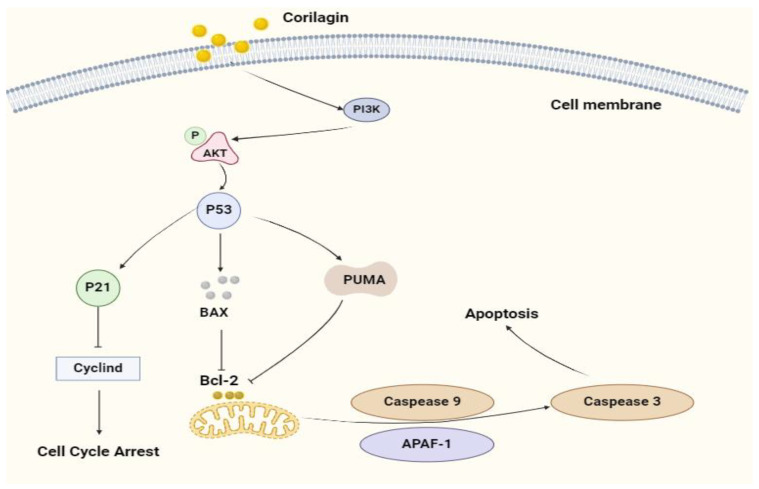
Corilagin-induced apoptosis signaling pathway in A2780 cells. Corilagin is effective against human ovarian cancer cell line A2780 cells in vitro by inhibiting the cell cycle and inducing apoptosis in tumor cells.

**Table 1 cimb-47-00105-t001:** PCR primer sequence.

Gene	Gene Forward Primer (5’→3’)	Reverse Primer (5’→3’)
*P53*	GTTCCGAGAGCTGAATGAGG	TCTGAGTCAGGCCCTTCTGT
*P21*	GACACCACTGGAGGGTGACT	CAGGTCCACATGGTCTTCCT
*CYCLIND*	AACTACCTGGACCGCTTCCT	CCACTTGAGCTTGTTCACCA
*BAX*	AAGAAGCTGAGCGAGTGTCT	GTTCTGATCAGTTCCGGCAC
*BCL-2*	GCCTTCTTTGAGTTCGGTGG	CAAATCAAACAGAGGCCGCA
*CASPEASE3*	ACTGGACTGTGGCATTGAGA	GCACAAAGCGACTGGATGAA
*CASPASE9*	GCCCCATATGATCGAGGACA	CAGAAACGAAGCCAGCATGT
*CYTOCHROME C*	ATGAAGTGTTCCCAGTGCCA	CTCTCCCCAGATGATGCCTT
*PERP*	TGCCATCATTCTCATTGCAT	AACCCCAGTTGAACTCATGG
*PUMA*	GAGGAGGAACAGTGGGCC	GGAGTCCCATGATGAGATTGT
*β-ACTIN*	CATCCGCAAAGACCTGTACG	CCTGCTTGCTGATCCACATC

**Table 2 cimb-47-00105-t002:** IC_50_ values of compounds on A2780 cells and IOSE-80 cells.

Cells	Compound	Time (h)	IC_50_ Value (μmol/mL)
A2780 Cells	Corilagin	24	61.32
		48	47.81
	5-Fu	24	36.46
		48	22.81
IOSE-80 Cells	Corilagin	24	263.72
		48	174.52

## Data Availability

The original contributions presented in the study are included in the article, further inquiries can be directed to the corresponding author.

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
