# Peer review of "Study on the Effects and Mechanism of Corilagin on A2780 Cell Apoptosis"

_cimb, 2025, doi:10.3390/cimb47020105_

Round 1

Reviewer 1 Report

Comments and Suggestions for Authors

In this work, authors have evaluated the potential of corilagin as a therapeutic agent for ovarian cancer, showing that the antiproliferative effect correlated with altered cell cycle, induction of apoptosis, differentially expressed apoptosis-related genes in corilagin-treated A2780 cells, primarily within the PI3K-AKT pathway.

The results were obtained with appropriate methods and seem to support the rationale of the study. Based on the findings, authors conclude that corilagin would have therapeutic potential for ovarian cancer A2780, especially through the PI3K/p53 pathway, claiming to have demonstrated for the first time the molecular mechanism of the effect of corilagin in these cells.

However, this statement would be stronger if similar results, at least some of them, were obtained on at least a second ovarian tumor cell line. Perhaps that is why in the last line of the abstract they need to specify the name of the cells and not as is usually done on the type of tumor in general.

Other points need to be better explained.

In Fig. 2, Table 2, Fig.4 the effect of corilagin is compared with 5FU, but not in the remaining experiments. The authors should explain this.

In addition, they should also explain why they chose 5FU as a reference drug, since it is not typically used to treat ovarian cancer. A Pt-derived drug would have been more appropriate, e.g. cisplatin or carboplatin.

Furthermore, the effect of 5FU could probably have been underestimated since, being an inhibitor of thymidylate synthase, it was used in conditions where the enzyme is less weakened since the cells were kept in DMEM medium which contains about 10 µM folic acid against physiological nanomolar values. Other media have approximately 4 times lower concentrations of folic acid, e.g. RPMI medium, about 2.2 µM.

The authors should also explain why different concentrations of corilagin were used for different times in the experiments in Figures 9 and 10.

It is usually better to keep concentrations and times homogeneous to better correlate the data from different experiments.

In the legend of Figure 11 it is mistakenly written gastric instead of ovarian cancer.

Reviewer 2 Report

Comments and Suggestions for Authors

This publication seems to be within the scope of journal. However it needs several corrections to be more acceptable for publication.

Were antibiotics used during A2780 culture? If they were used, please add this information to the culture conditions.

Line 34: Please add “(β-1-O-galloyl-3,6-(R)-hexahydroxydiphenoyl-D-glucose)”.

Line 68: it should be “Cell Counting Kit-8 (CCK-8)” instead of “CCK-8 reagent”.

Line 82: please add the composition of the DMEM medium.

Line 16, 244, 247, 249, 411, 462: It should be ”G0/G1 and G2/M” instead of ”G0/G1 and G2/M”.

Line 21: It should be ”p53 and Bax genes” instead of ”p53 and Bax genes”.

Line 34: Please include the reference to Fig. 1 in your text.

Line 37: It should be ”emblica” instead of ”emblica”. Please add Latin names of all plants.

Line 84: It should be ”CO2” instead of ”CO2”.

Line 96, 222, 223, 224 and others: It should be ”IC50” instead of ”IC50”. Please check carefully the whole manuscript and correct evident mistake.

Line 108: It should be ”propidium iodide” instead of ”Propidium iodide”. The names of chemical compounds should begin with a regular letter, not a capital letter. Please check carefully the whole manuscript and correct evident mistake.

Line 109 please add information about concentration of PI.

Line 153, 188: It should be ”1×105 cells/mL” instead of ”1×105/mL”.

Line 191, 192: It should be ”mL” instead of ”ml”.

Line 426 and in the title of figure 5: It should be ”Ca2+” instead of ”Ca2+”.

Round 2

Reviewer 1 Report

Comments and Suggestions for Authors

The authors have overall answered all the questions posed exhaustively. Although in the answer to the first question, there would be a rather important piece of data regarding the selectivity of Corilagin which is very little toxic against human normal ovarian cell (IOSE-80). Perhaps it is worth mentioning it in the text already in this work and not only in a possible future work, as they stated in the answer.
